# Test-Time Amendment with a Coarse Classifier for Fine-Grained Classification

**Kanishk Jain**[1]**, Shyamgopal Karthik**[2]**, Vineet Gandhi**[1]
[1]IIIT Hyderabad [2]University of Tübingen

## Abstract

We investigate the problem of reducing mistake severity for fine-grained classification. Fine-grained classification can be challenging, mainly due to the requirement of domain expertise for accurate annotation. However, humans are particularly adept at performing coarse classification as it requires relatively low levels of expertise. To this end, we present a novel approach for Post-Hoc Correction called Hierarchical Ensembles (HiE) that utilizes label hierarchy to improve the performance of fine-grained classification at test-time using the coarse-grained predictions. By only requiring the parents of leaf nodes, our method significantly reduces *avg. mistake severity* while improving *top-1* accuracy on the *iNaturalist-19* and *tieredImageNet-H* datasets, achieving a new state-of-the-art on both benchmarks. We also investigate the efficacy of our approach in the semi-supervised setting. Our approach brings notable gains in *top-1* accuracy while significantly decreasing the severity of mistakes as training data decreases for the fine-grained classes. The simplicity and post-hoc nature of HiE renders it practical to be used with any off-the-shelf trained model to improve its predictions further. The code is available at: `https://github.com/kanji95/Hierarchical-Ensembles`

## 1  Introduction

Over the past decade, large-scale datasets have been instrumental in driving the rapid progress of computer vision/image recognition. However, in settings that require experts to annotate samples, collecting large amounts of labeled data can be prohibitively expensive. *Fine-Grained Visual Classification (FGVC)* is one such example, where one would need a domain expert to be able to identify the category for a particular sample. However, it can be cheaper to obtain coarse labels for the same samples in these settings. For instance, in Figure 1 while it is evident to spot and identify a butterfly or a bird, one would need a *Lepidopterist* or an *Ornithologists* to categorize them as *Junonia Genoveva* or *Water Ouzel*.

Therefore, one popular research direction in recent times has been to utilize the availability of a larger set of images with coarse labels to improve the performance (i.e *top-1* accuracies) of the neural network models on various fine-grained classification benchmarks [1, 2]. Using coarse labels in a classification setting introduces the concept of a label hierarchy, where all the labels in a dataset are connected through some taxonomy. This taxonomy could either be defined from the biological taxonomy, as with many species recognition datasets or could be derived from language-based ontologies such as WordNet [3]. Another mainstream usage of label hierarchies in recent times is to use them to reduce the severity of mistakes committed by various classification models [4, 5, 6, 7, 8]. For instance, mistaking a car for a bus is a better mistake than mistaking a car for a lamppost. This originated from the extensive work on cost-sensitive classification [9, 10]. However, defining pairwise costs can be a tedious process and scales quadratically with the number of classes. Therefore, the recent works have all focused on using the label hierarchy and defining costs automatically based on graph distances (e.g. using the height of the least common ancestor between the predicted and the ground truth class as a proxy for cost). The research in this direction aims to invent methods that at

37th Conference on Neural Information Processing Systems (NeurIPS 2023).

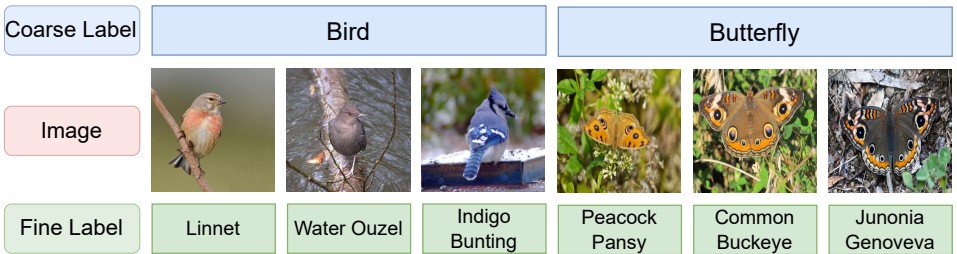

Figure 1: While it is straightforward to identify the images based on their coarse labels, it is challenging to distinguish them based on fine-grained labels without domain expertise.

least retain the accuracies of the backbone baseline models while reducing the severity of mistakes committed by it.

While these two research directions might seem entirely different, they work with the same inputs (i.e., images, labels, and the label hierarchy). Therefore, one would ideally want a method that can use additional coarsely labelled samples to improve the model's accuracy and utilise the label hierarchy to reduce the severity of the mistakes. However, these two goals have been tackled independently. Post-hoc correction methods [4, 7], learning hierarchy-aware features [6, 8, 11] as well as structured embedding spaces [5, 12] have been proposed to reduce the severity of mistakes. Coarse labels have been used for self-training/pseudo-labelling [13] and with an additional objective to regularise/enhance the training [2] in the semi-supervised learning frameworks.

In this work, we propose a simple method that is able to bring significant improvements to both of these problems. Our fundamental insight is that models trained for different granularities focus on different aspects. This is also corroborated by recent work [11, 8], which shows that when training a model jointly, coarse-level label prediction exacerbates fine-grained feature learning. Therefore, we train two separate models for the coarse and fine-grained labels. At inference, we employ a modified decision rule, which computes the argmax over the normalized score of coarse and fine grained predictions. We refer to the proposed strategy as a Hierarchical Ensemble (HiE). We find that employing HiE not only reduces the severity of mistakes, but is able to improve *top-1* accuracies, which are far more pronounced in the semi-supervised setting. More formally, we make the following contributions:

1. We introduce a simple post-hoc mechanism called Hierarchical Ensembles for incorporating cues from multiple models trained for different granularities of a hierarchy tree.

2. We demonstrate that HiE brings considerable reductions in mistake severity while improving the *top-1 accuracy*. HiE significantly outperforms two competent baselines and six other methods from the prior art and achieves state-of-the-art performance on both *iNaturalist-19* and *tieredImageNet-H* benchmarks.

3. We illustrate that HiE brings consistent gains when plugged with existing semi-supervised methods exploiting additional coarse labeled data to improve the fine-grained classification. On *iNaturalist-19* dataset, HiE combined with Moco [14] and hierarchical loss [2], recovers 86% of the underlying performance while using only 0.5% of the fine-grained annotations.

## 2   Related Work

**Learning from Taxonomic Labels:** There has been a long history of utilizing label hierarchies for better classification performance. Sila and Freitas [15] perform a comprehensive survey of hierarchical classification methods applied to various tasks in different application domains. Broadly, the methods that train hierarchy-aware features can be categorized into three categories a) *Label-Embedding methods*, b) *Hierarchical architectures*, and c) *Hierarchical loss functions*.

The label-embedding methods project the categories into a semantic embedding space [12, 16] instead of viewing them as one-hot encodings and are widely used, especially in zero-shot learning [17]. The semantic embedding space can be derived either from side-information such as class attributes [16],

taxonomies [6], or from natural language [12]. These can be further enhanced by learning these embeddings on a hypersphere [5] or hyperbolic spaces [18, 19, 20].

On the other hand, hierarchical loss functions try to modify the training objective to incorporate the label hierarchy [21, 6, 22, 8, 23]. The most common approach here is to introduce a loss at every level in the hierarchy [21, 6, 8]. The challenge in this direction has been to ensure that the coarse-grained task does not hamper the accuracy of the fine-grained classifier [11, 8].

There have also been attempts to modify the architecture based on the label hierarchy. For instance, popular ideas revolve around having branches at different layers of the model [24], predicting the conditional probabilities at each node in the hierarchy [25, 26], or partitioning the feature space using the levels in the hierarchies [11].

Most closely related to our work are the works that attempt to incorporate the label hierarchy information post-hoc during inference. Here, traditional Conditional Risk Minimization (CRM) [27] was applied in this setting by [4] and recently revisited by [7]. Our proposed approach is also applied during inference; however, it is orthogonal to CRM, and we demonstrate that CRM can be used to improve our results further. Concurrent to our work, [28] proposed a similar approach to utilize subclasses to improve superclass recognition performance in a post-hoc manner with vision-language models [29].

**Fine-Grained Visual Classification:** Contrary to the popular image recognition benchmarks, fine-grained visual classification focuses on recognizing the differences between similar-looking categories [30, 31, 32]. Here, methods often focus on learning discriminative features that use local information since most categories share a similar global structure [33, 34, 35]. However, the most related to our work are the ones that use class taxonomies to improve fine-grained classification performance. This is most frequently done in the semi-supervised setting [1], where one has access to an additional set of weakly labeled samples. Here, approaches have explored incorporating a hierarchical loss on the coarse labels [2] or using the coarse labels to filter pseudo-labels [13] in addition to the standard semi-supervised learning techniques of consistency regularization [36], pseudo-labeling [37, 38], and contrastive learning [14]. Our proposed approach is complementary to these works and can be applied to all of these works to bring additional improvements.

## 3 Methodology

### 3.1 Problem Formulation

We consider the fine-grained classification task with a label hierarchy $\mathcal{H}$ of $L$ levels defined over the class labels. The leaf nodes of $\mathcal{H}$ correspond to fine-grained classes at level $L$, and the class labels at level $l < L$ are considered coarse-grained labels. The goal is to use the information from coarse annotations to improve the performance on fine labels. Formally, given a dataset $\mathcal{X} = \{(x_i, y_i^l) \mid i = 1, 2, \ldots, N\}$ consisting of $N$ training images and their respective ground truth labels at level $l$, where label $y_i^l \in \mathcal{Y}^l = \{1, 2, ..., N_l\}$, and $N_l$ corresponds to the number of classes at level $l$, the task is to train a classifier $f_\theta^L : \mathcal{X} \to p(\mathcal{Y}^L)$, on fine-grained class labels by making use of the information available through coarse-grained labels $\mathcal{Y}^{l<L}$.

### 3.2 Proposed Method

Suppose we have two classifiers, $f_\theta^L$ and $f_\phi^{L-1}$ trained on hierarchy levels $L, L-1$ with $\theta$ and $\phi$ being their parameters, respectively, then for a sample $x \sim D$, we have, $\hat{y}_L = f_\theta^L(x)$ and $\hat{y}_{L-1} = f_\phi^{L-1}(x)$ as logits of two classifiers for the input $x$. We further define:

$$Q = [q_1, q_2, ..., q_{N_L}] = \text{softmax}(\hat{y}_L)$$

$$R = [r_1, r_2, ..., r_{N_{L-1}}] = \text{softmax}(\hat{y}_{L-1})$$

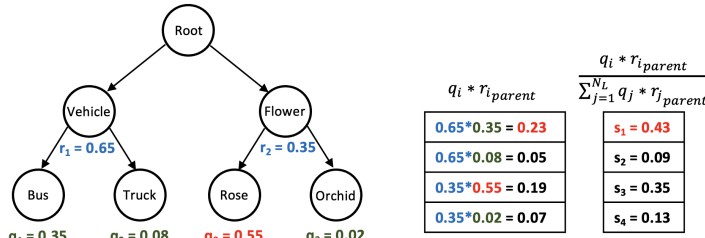

Figure 2: Consider a four class hierarchy tree and corresponding fine-grained predictions ($q_i$) and coarse level predictions ($r_i$) obtained using independent models. Our work aims to improve the fine grained classification by leveraging the predictions at the coarse level. In the given example, the class prediction changes from 'rose' to 'bus' after the post-hoc correction.

We reweigh the fine-grained predictions $\mathcal{Q}$, given the softmax probabilities $\mathcal{R}$ obtained from the coarse-grained classifier:

$$P(i|x,\theta,\phi,\mathcal{H}) = \frac{P(i|x;\theta)P(i_{parent}|x;\phi)}{\sum_{j=1}^{N_L} P(j|x;\theta)P(j_{parent}|x;\phi)}, \quad \text{where } i_{parent} = \mathcal{H}(i) \tag{1}$$

The denominator is the normalization term, ensuring $\sum_{i=1}^{N_L} P(i|x,\mathcal{H}) = 1$. In the studied setting, $P(i|x,\mathcal{H})$ simplifies to:

$$P(i|x,\mathcal{H}) = s_i = \frac{q_i \cdot r_{i_{parent}}}{\sum_{j=1}^{N_L} q_j \cdot r_{j_{parent}}} \tag{2}$$

The main modification we make in the classification strategy at the fine grained level is in the decision rule, which now selects the class that maximizes the normalized score:

$$\underset{i}{\operatorname{argmax}} \, P(i|x,\mathcal{H}) \tag{3}$$

In Figure 2, we illustrate a four-class example, comparing standard cross-entropy predictions with those from our proposed method. We would like to point out that the proposed technique can be interpreted as a Hierarchical Ensemble. Existing literature in neural networks-based ensemble learning has often focused on classifiers trained at a single granularity [39] and has largely overlooked ensembles with hierarchical context. In this work, we highlight the practical benefits of hierarchical ensembles for improving the fine-grained classification task by choosing the simplest possible method of multiplying the predictions at different levels of hierarchy.

We show that if we make a correct prediction at the coarse level, the proposed Hierarchical Ensemble (HiE) is guaranteed to improve the downstream predictions at the fine-grained classification task.

**Theorem 3.1.** *Assuming, $Q = [q_1, q_2, ..., q_{N_L}]$ and $R = [r_1, r_2, ..., r_{N_{L-1}}]$ are the predictions obtained at the fine and coarse grained labels for a given input $x$, such that $\sum_{i=1}^{N_L} q_i = 1$ and $\sum_{i=1}^{N_{L-1}} r_i = 1$. For the ground truth labels $g$ and $g_{parent}$ at the fine grained and the coarse grained levels respectively:*

*Now assuming that the coarse label is correctly predicted by the coarse prediction network i.e. $\operatorname{argmax}(R) = g_{parent}$, we wish to prove that:*

$$\frac{q_g \cdot r_{g_{parent}}}{\sum_{j=1}^{N_L} q_j \cdot r_{j_{parent}}} \geq q_g.$$

*Proof.* The denominator iterates over the fine grained predictions and multiplies them with their parent's prediction scores. This is equivalent to iterating over the coarse label predictions and multiplying with the sum of prediction scores for all its children. By rewriting the denominator, we obtain:

$$\sum_{j=1}^{N_L} q_j \cdot r_{j_{parent}} = \sum_{j=1}^{N_{L-1}} r_j \sum_{i \in j_{child}} q_i$$

Assuming, $\sum_{i \in j_{child}} q_i = z_j$

$$\sum_{j=1}^{N_{L-1}} r_j \sum_{i \in j_{child}} q_i = \sum_{j=1}^{N_{L-1}} r_j \cdot z_j = R^T Z$$

Invoking Holder's inequality, using $a = \infty$ and $b = 1$, $(1/a + 1/b = 1)$, we obtain:

$$R^T Z \leq \|R\|_\infty \|Z\|_1$$

Since, $\|R\|_\infty = \max(R) = r_{g_{parent}}$ and $\|Z\|_1 = \sum_{i=1}^{N_L} q_i = 1$. We can say:

$$R^T Z \leq \|R\|_\infty \|Z\|_1 \leq r_{g_{parent}}$$

Given the above equation, we can conclude that:

$$\frac{r_{g_{parent}}}{\sum_{j=1}^{N_L} q_j \cdot r_{j_{parent}}} \geq 1.$$

$\square$

When correct predictions are made at both the coarse and fine-grained levels, the method will help increase the confidence of the prediction. A more interesting case occurs when the coarse-level prediction is correct; however, the fine-grained classifier makes a wrong prediction. The hierarchical ensemble can shift the prediction to the correct sub-tree, reducing the mistake severity and potentially improving the *top-1* accuracy.

## 4   Experiments & Results

Like prior works [6, 8, 7], we evaluate our approach on the *iNaturalist-19* [40] and *tieredImageNet-H* [41] datasets. The *iNaturalist-19* dataset is a species classification dataset containing plants, animals, fungi, etc., and has an 8-level hierarchy. The *tieredImageNet-H* dataset contains classes from a wide range of categories, including musical instruments, animal breeds, tools, and plant species. It has a 13-level hierarchy derived from the WordNet hierarchy. However, rather than using the full hierarchy of categories, for our method, we only use the leaf classes and their parent categories during training and inference. The *iNaturalist-19* dataset includes 1010 fine-grained "species" classes and 72 coarse-grained classes for the "genus" taxonomy. The *tieredImageNet-H* dataset has 608 leaf classes and 201 parent classes. We train classifiers for each level independently, using the same hyperparameter settings.

In addition, we also evaluate the performance of our approach in a semi-supervised setting. Specifically, we want to test the performance on fine-grained classification tasks using a large number of coarsely annotated examples and a limited number of finely annotated examples. We use the *iNaturalist-19* dataset for this setting; the coarse labels for all the images are used, while for fine-grained labels, we sample a limited number of images per class label during training. We used the "species" and "genus" taxonomies for the fine and coarse level labels and performed experiments for cases when the number of images per fine label is 100, 50, 25, and 10. Similar to the supervised setting, we train separate classifiers on coarse and fine labels.

To align with prior efforts [7, 8], all of our experiments are performed using the ResNet-18 backbone. We randomly crop a portion of images and resize them to $224 \times 224$ resolution for both datasets. For the *iNaturalist-19* dataset, we initialize the ResNet-18 weights with a pre-trained ImageNet model and train the classifiers using a customized SGD optimizer for 100 epochs, with different learning rates for the backbone network and the fully connected layer (0.01 and 0.1, respectively). For the *tieredImageNet-H* dataset, we train the ResNet-18 from scratch (since it is derived from the ImageNet

| Method | Top-1 Error(↓) | Mistakes severity(↓) | Hier dist@1(↓) | Hier dist@5(↓) | Hier dist@20(↓) |
|---|---|---|---|---|---|
| | | | Without CRM | | |
| Cross-Entropy | 36.44 ± 0.061 | 2.39 ± 0.007 | 0.87 ± 0.004 | 1.97 ± 0.002 | 3.25 ± 0.002 |
| Cross-Entropy-H | 37.63 ± 0.054 | 2.37 ± 0.005 | 0.89 ± 0.009 | 1.86 ± 0.001 | 2.96 ± 0.001 |
| HiE-Self | 36.36 ± 0.047 | 2.36 ± 0.014 | 0.86 ± 0.005 | 1.44 ± 0.003 | 2.31 ± 0.004 |
| Barz & Denzler [5] | 62.63 ± 0.278 | 1.99 ± 0.008 | 1.24 ± 0.005 | 1.49 ± 0.005 | 1.97 ± 0.005 |
| YOLO-v2 [25] | 44.37 ± 0.106 | 2.42 ± 0.003 | 1.08 ± 0.004 | 1.90 ± 0.003 | 2.87 ± 0.010 |
| HXE $\alpha$=0.1 [6] | 41.48 ± 0.204 | 2.41 ± 0.009 | 1.00 ± 0.006 | 1.77 ± 0.011 | 2.69 ± 0.021 |
| HXE $\alpha$=0.6 [6] | 45.45 ± 0.014 | 2.24 ± 0.006 | 1.02 ± 0.003 | 1.70 ± 0.005 | 2.55 ± 0.005 |
| Soft-labels $\beta = 30$ [6] | 41.67 ± 0.134 | 2.32 ± 0.010 | 0.97 ± 0.006 | 1.50 ± 0.006 | 2.23 ± 0.005 |
| Soft-labels $\beta = 4$ [6] | 74.70 ± 0.212 | 1.82 ± 0.005 | 1.36 ± 0.004 | 1.49 ± 0.003 | 1.96 ± 0.004 |
| Chang et al. [11] | 37.23 ± 0.175 | 2.28 ± 0.006 | 0.85 ± 0.004 | 1.75 ± 0.005 | 3.02 ± 0.008 |
| HAF [8] | 36.40 ± 0.092 | 2.28 ± 0.012 | 0.83 ± 0.002 | 1.62 ± 0.002 | 2.55 ± 0.003 |
| HiE (Ours) | 35.33 ± 0.037 | 2.15 ± 0.003 | 0.76 ± 0.001 | 1.33 ± 0.001 | 2.19 ± 0.003 |
| | | | With CRM | | |
| Cross-Entropy [7] | 36.51 ± 0.083 | 2.33 ± 0.001 | 0.85 ± 0.002 | 1.32 ± 0.001 | 1.86 ± 0.002 |
| Cross-Entropy-H | 37.70 ± 0.038 | 2.33 ± 0.011 | 0.88 ± 0.003 | 1.33 ± 0.002 | 1.87 ± 0.004 |
| HiE-Self | 36.45 ± 0.057 | 2.34 ± 0.019 | 0.85 ± 0.003 | 1.32 ± 0.003 | 1.86 ± 0.008 |
| Barz & Denzler | 75.61 ± 0.098 | 4.62 ± 0.032 | 3.49 ± 0.025 | 3.66 ± 0.024 | 3.84 ± 0.018 |
| YOLO-v2 | 45.17 ± 0.046 | 2.43 ± 0.001 | 1.10 ± 0.001 | 1.50 ± 0.001 | 1.99 ± 0.002 |
| HXE $\alpha$=0.1 | 41.47 ± 0.220 | 2.38 ± 0.011 | 0.99 ± 0.008 | 1.41 ± 0.006 | 1.93 ± 0.005 |
| HXE $\alpha$=0.6 | 45.60 ± 0.017 | 2.21 ± 0.008 | 1.01 ± 0.003 | 1.40 ± 0.004 | 1.40 ± 0.004 |
| Soft-labels $\beta = 30$ | 41.99 ± 0.126 | 2.31 ± 0.009 | 0.97 ± 0.007 | 1.40 ± 0.005 | 1.91 ± 0.005 |
| Soft-labels $\beta = 4$ | 77.34 ± 0.262 | 2.06 ± 0.012 | 1.60 ± 0.007 | 1.72 ± 0.008 | 2.14 ± 0.007 |
| Chang et al. [11] | 37.31 ± 0.145 | 2.24 ± 0.008 | 0.84 ± 0.002 | 1.30 ± 0.002 | 1.84 ± 0.002 |
| HAF [8] | 36.48 ± 0.095 | 2.25 ± 0.012 | 0.82 ± 0.003 | 1.29 ± 0.004 | 1.84 ± 0.002 |
| HiE (Ours) | 35.42 ± 0.025 | 2.14 ± 0.015 | 0.75 ± 0.005 | 1.23 ± 0.005 | 1.79 ± 0.005 |

Table 1: Results comparing *top-1* error(%) and hierarchical metrics for *iNaturalist-19*. Results in the *Top* block are reported without using CRM [7] technique and *Bottom* block are reported using CRM. Methods highlighted with aqua green are the best performing methods in top-1 error (%). We use dark green for the top performers and light green for the runners-up in each metric.

dataset) and use the AMSGrad variant of the Adam optimizer with a learning rate of $1e^{-5}$ for 120 epochs. We conduct five runs for each experiment in tables 1 and 2 and report the mean and standard deviations.

**Evaluation Metrics:** Like previous works [6, 7], we validate our approach using the *top-1* accuracy, *avg. mistake severity* and *hierarchical distance@k* metrics. *Avg. mistake severity* metric computes the average height of the lowest common ancestor (LCA) between the predicted class and the ground truth class for the incorrectly classified samples. Since this metric considers only the incorrect predictions, it is possible to reduce the mistake severity by making more low cost mistakes [7], as a consequence of which the *top-1* accuracy decreases. To overcome this issue, *hierarchical distance@k* is used, which measures the average height of *top-k* predictions with the ground truth class for all samples. For $k > 1$, this metric captures the quality of ranking imposed by the network in the label space.

## 4.1 Making better mistakes

In this section, we demonstrate that the proposed hierarchical ensembles significantly reduce the severity of mistakes compared to the current state-of-the-art methods. We compare our method with [5], YOLO-v2 [25]; both approaches proposed by [6] with varying parameters - soft-labels and HXE. We also compare against the method proposed by [11] for multi-granular classification and HAF [8] imposing consistency on prediction heads across label hierarchy. We contrast the performance of the methods mentioned above to the cross-entropy baseline.

We also propose two custom baselines called Cross-Entropy-H and HiE-Self. Cross-Entropy-H obtains coarse class predictions by marginalizing the prediction of its fine-grained children classes. It then employs an additional hierarchical loss term [2] over the obtained coarse predictions. HiE-Self applies a hierarchical ensemble using the coarse predictions obtained post marginalization over fine-grained classes (instead of training a separate network for the coarse level).

We also experiment with the CRM technique proposed by [7]. Since CRM is a post-hoc approach that reweighs the probability distribution of samples obtained from any trained model, it can be applied to

pre-trained models of all the approaches. Therefore, in each of the Tables 1-2, we group the results to report evaluation metrics with and without using CRM at test time.

| Method | Top-1 error(↓) | Mistakes severity(↓) | Hier dist@1(↓) | Hier dist@5(↓) | Hier dist@20(↓) |
|---|---|---|---|---|---|
| | | | Without CRM | | |
| Cross-Entropy | 30.64 ± 0.030 | 7.07 ± 0.010 | 2.17 ± 0.006 | 5.70 ± 0.003 | 7.25 ± 0.003 |
| Cross-Entropy-H | 32.87 ± 0.042 | 7.13 ± 0.031 | 2.35 ± 0.003 | 5.70 ± 0.009 | 7.14 ± 0.012 |
| HiE-Self | 30.78 ± 0.054 | 7.05 ± 0.028 | 2.19 ± 0.004 | 5.35 ± 0.009 | 6.92 ± 0.007 |
| Barz & Denzler [5] | 39.73 ± 0.240 | 6.80 ± 0.019 | 2.70 ± 0.022 | 5.48 ± 0.271 | 6.21 ± 0.005 |
| YOLO-v2 [25] | 33.37 ± 0.082 | 7.02 ± 0.004 | 2.34 ± 0.016 | 5.85 ± 0.011 | 7.43 ± 0.016 |
| HXE $\alpha$=0.1 [6] | 30.72 ± 0.036 | 7.00 ± 0.019 | 2.15 ± 0.005 | 5.62 ± 0.008 | 7.08 ± 0.015 |
| HXE $\alpha$=0.6 [6] | 34.50 ± 0.007 | 6.73 ± 0.014 | 2.32 ± 0.003 | 5.48 ± 0.001 | 6.78 ± 0.003 |
| Soft-labels $\beta = 30$ [6] | 30.53 ± 0.194 | 7.05 ± 0.009 | 2.15 ± 0.013 | 5.66 ± 0.002 | 7.14 ± 0.008 |
| Soft-labels $\beta = 4$ [6] | 38.99 ± 0.105 | 6.60 ± 0.024 | 2.57 ± 0.004 | 5.13 ± 0.002 | 6.21 ± 0.001 |
| Chang et al. [11] | 33.46 ± 0.026 | 6.99 ± 0.010 | 2.34 ± 0.006 | 5.75 ± 0.005 | 7.34 ± 0.010 |
| HAF [8] | 30.50 ± 0.010 | 7.03 ± 0.024 | 2.14 ± 0.008 | 5.62 ± 0.011 | 6.99 ± 0.009 |
| HiE (Ours) | 29.81 ± 0.081 | 6.95 ± 0.013 | 2.07 ± 0.014 | 5.30 ± 0.001 | 6.86 ± 0.001 |
| | | | With CRM | | |
| Cross-Entropy [7] | 30.56 ± 0.020 | 7.01 ± 0.007 | 2.14 ± 0.006 | 4.93 ± 0.002 | 6.11 ± 0.001 |
| Cross-Entropy-H | 32.93 ± 0.029 | 7.05 ± 0.006 | 2.32 ± 0.004 | 5.01 ± 0.008 | 6.14 ± 0.003 |
| HiE-Self | 30.79 ± 0.031 | 7.04 ± 0.008 | 2.17 ± 0.004 | 4.98 ± 0.005 | 6.14 ± 0.002 |
| Barz & Denzler | 83.55 ± 0.000 | 11.94 ± 0.000 | 11.92 ± 0.000 | 11.91 ± 0.000 | 11.91 ± 0.000 |
| YOLO-v2 | 33.98 ± 0.099 | 6.99 ± 0.011 | 2.38 ± 0.012 | 5.05 ± 0.001 | 6.17 ± 0.001 |
| HXE $\alpha$=0.1 | 30.80 ± 0.079 | 6.95 ± 0.021 | 2.14 ± 0.005 | 4.94 ± 0.003 | 6.11 ± 0.002 |
| HXE $\alpha$=0.6 | 34.68 ± 0.003 | 6.69 ± 0.007 | 2.32 ± 0.001 | 4.99 ± 0.005 | 6.13 ± 0.003 |
| Soft-labels $\beta = 30$ | 30.69 ± 0.125 | 6.99 ± 0.007 | 2.15 ± 0.008 | 4.95 ± 0.001 | 6.11 ± 0.001 |
| Soft-labels $\beta = 4$ | 82.72 ± 0.079 | 7.54 ± 0.001 | 6.24 ± 0.005 | 6.94 ± 0.005 | 7.25 ± 0.002 |
| Chang et al. [11] | 33.73 ± 0.033 | 6.93 ± 0.015 | 2.34 ± 0.002 | 5.02 ± 0.007 | 6.15 ± 0.001 |
| HAF [8] | 30.63 ± 0.007 | 6.97 ± 0.024 | 2.14 ± 0.008 | 4.95 ± 0.004 | 6.11 ± 0.001 |
| HiE (Ours) | 29.89 ± 0.082 | 6.93 ± 0.013 | 2.07 ± 0.013 | 4.93 ± 0.001 | 6.11 ± 0.001 |

Table 2: Results comparing top-1 error(%) and hierarchical metrics on the test set of *tieredImageNet-H*. The *Top* block reports results without using CRM [7] and the *Bottom* block are reported using CRM. Methods highlighted with aqua green are the best performing methods in top-1 error (%). We use dark green for the top performers and light green for the runners-up in each metric.

Apart from HAF, all the other methods from the prior art trade-off the *top-1* accuracy with the mistake severity (Tables 1&2). The results contrast with the problem's main objective, i.e., improve the hierarchical metrics by maintaining or improving the *top-1* error. In some of the instances (e.g. HXE $\alpha = 0.6$, Soft-labels $\beta = 30$), the *avg. mistake severity* is reduced by making additional low-severity mistakes (indicated by an increase in *top-1* error and *Hier dist@1*). The multi-head approach by [11] brings minor gains, and HAF further improves upon it by introducing consistency loss among different head predictions.

The proposed minimalist approach of Hierarchical Ensembles (HiE) significantly outperforms all the methods; on both datasets, with and without CRM. On the three metrics of *top-1* error, *avg. mistake severity*, and *Hier dist@1*, the performance achieved by *HiE without CRM* betters all other methods post-CRM. The gains achieved by HiE on *Hier dist@5* and *Hier dist@20* metrics are more pronounced, for instance, improving over HAF without CRM on *iNaturalist-19* dataset by 17% and 14% respectively. These two ranking metrics compare the ordering of the classes provided by each of these classifiers. The results illustrate that HiE is able to reliably align the predictions with the hierarchy. The results are interesting given that HiE only employs the bottom two levels of the hierarchy, in contrast to other methods which utilize the entire hierarchy tree.

Compared with the baselines, on the *iNaturalist-19* dataset, the Cross-Entropy-H brings minor gains on *Hier dist@5* and *Hier dist@20* metrics; however, the *top-1* error increases by over a percent. On *tieredImageNet-H*, Cross-Entropy-H fails to give any improvements. In contrast, the HiE-Self baseline is able to retain the original *top-1* performance while bringing noticeable gains on *Hier dist@k* metrics. It illustrates that hierarchical ensembles are helpful even when applied on a single network. However, training separate networks for coarse and fine-grained levels allows us to learn complementary features which are then combined using our HiE approach.

| Methods | #Imgs/Label | without HiE | | | | | | with HiE | | | | | |
|---|---|---|---|---|---|---|---|---|---|---|---|---|---|
| | | Top-1 Acc | | Avg. Mistake Severity | | Hierarchical Distance@1 | | Top-1 Acc | | Avg. Mistake Severity | | Hierarchical Distance@1 | |
| | | w/o CRM | w CRM | w/o CRM | w CRM | w/o CRM | w CRM | w/o CRM | w CRM | w/o CRM | w CRM | w/o CRM | w CRM |
| CrossEntropy | | 53.90 | 53.93 | 2.55 | 2.48 | 1.17 | 1.14 | 56.95 | 56.82 | 2.07 | 2.06 | 0.89 | 0.89 |
| Pseudo-Label | | 51.25 | 51.09 | 2.52 | 2.46 | 1.23 | 1.20 | 54.78 | 54.61 | 2.04 | **2.03** | 0.92 | 0.92 |
| MoCo | | 54.72 | 54.51 | 2.41 | 2.36 | 1.09 | 1.07 | 57.07 | 56.91 | 2.05 | 2.04 | 0.88 | 0.88 |
| MocoST | | 55.77 | 55.59 | 2.42 | 2.37 | 1.07 | 1.05 | 57.99 | 57.78 | 2.07 | 2.06 | 0.87 | 0.87 |
| CrossEntropy-H | 100 | 55.76 | 55.71 | 2.48 | 2.42 | 1.10 | 1.07 | 58.50 | 58.32 | 2.08 | 2.07 | **0.86** | **0.86** |
| PseudoLabel-H | | 56.18 | 55.98 | 2.47 | 2.41 | 1.08 | 1.06 | **58.80** | 58.79 | 2.09 | 2.08 | **0.86** | **0.86** |
| ST-H | | 54.82 | 54.71 | 2.47 | 2.41 | 1.12 | 1.09 | 57.51 | 57.45 | 2.07 | 2.07 | 0.88 | 0.88 |
| Moco-H | | 56.05 | 55.85 | 2.47 | 2.42 | 1.09 | 1.07 | 58.56 | 58.49 | 2.09 | 2.09 | 0.87 | 0.87 |
| MocoST-H | | 55.74 | 55.49 | 2.47 | 2.41 | 1.10 | 1.07 | 58.32 | 58.17 | 2.08 | 2.08 | 0.87 | 0.87 |
| CrossEntropy | | 45.47 | 45.60 | 2.70 | 2.61 | 1.47 | 1.42 | 50.39 | 50.29 | 2.00 | 1.99 | 0.99 | 0.99 |
| Pseudo-Label | | 45.97 | 45.84 | 2.57 | 2.51 | 1.39 | 1.36 | 50.59 | 50.43 | 2.01 | **1.99** | 0.98 | 0.98 |
| MoCo | | 47.94 | 47.75 | 2.55 | 2.50 | 1.33 | 1.31 | 51.72 | 51.61 | 2.01 | 2.00 | 0.97 | 0.97 |
| MocoST | | 49.31 | 49.20 | 2.54 | 2.48 | 1.29 | 1.26 | 52.87 | 52.73 | 2.02 | 2.00 | 0.95 | 0.95 |
| CrossEntropy-H | 50 | 54.87 | 54.83 | 2.50 | 2.43 | 1.13 | 1.10 | 57.69 | 57.50 | 2.08 | 2.07 | 0.88 | 0.88 |
| PseudoLabel-H | | 55.17 | 54.97 | 2.48 | 2.40 | 1.11 | 1.08 | **57.97** | 57.84 | 2.07 | 2.07 | **0.87** | **0.87** |
| ST-H | | 52.42 | 52.27 | 2.50 | 2.43 | 1.19 | 1.16 | 55.74 | 55.65 | 2.04 | 2.05 | 0.90 | 0.91 |
| Moco-H | | 54.76 | 54.58 | 2.48 | 2.42 | 1.12 | 1.10 | 57.65 | 57.45 | 2.07 | 2.08 | 0.88 | 0.88 |
| MocoST-H | | 54.53 | 54.26 | 2.48 | 2.42 | 1.13 | 1.11 | 57.37 | 57.21 | 2.07 | 2.08 | 0.88 | 0.89 |
| CrossEntropy | | 36.47 | 36.63 | 2.87 | 2.77 | 1.82 | 1.76 | 43.28 | 43.22 | 1.93 | **1.91** | 1.10 | 1.09 |
| Pseudo-Label | | 39.13 | 39.20 | 2.72 | 2.66 | 1.66 | 1.61 | 44.96 | 44.91 | 1.95 | 1.93 | 1.07 | 1.06 |
| MoCo | | 39.48 | 39.55 | 2.74 | 2.68 | 1.66 | 1.62 | 45.06 | 45.00 | 1.94 | 1.92 | 1.06 | 1.06 |
| MocoST | | 41.22 | 41.12 | 2.72 | 2.63 | 1.60 | 1.55 | 46.37 | 46.30 | 1.96 | 1.95 | 1.05 | 1.05 |
| CrossEntropy-H | 25 | 53.26 | 53.18 | 2.51 | 2.44 | 1.17 | 1.14 | 56.43 | 56.21 | 2.06 | 2.05 | 0.90 | 0.90 |
| PseudoLabel-H | | 53.30 | 53.22 | 2.51 | 2.43 | 1.17 | 1.14 | **56.57** | 56.44 | 2.06 | 2.05 | **0.89** | **0.89** |
| ST-H | | 48.29 | 48.19 | 2.59 | 2.52 | 1.34 | 1.30 | 52.69 | 52.52 | 2.01 | 2.01 | 0.95 | 0.96 |
| Moco-H | | 53.01 | 52.97 | 2.50 | 2.43 | 1.17 | 1.14 | 56.29 | 56.25 | 2.06 | 2.05 | 0.90 | 0.90 |
| MocoST-H | | 53.21 | 53.02 | 2.49 | 2.44 | 1.17 | 1.14 | 56.34 | 56.12 | 2.05 | 2.06 | 0.90 | 0.90 |
| CrossEntropy | | 24.79 | 25.17 | 3.12 | 2.98 | 2.34 | 2.23 | 33.33 | 33.32 | 1.82 | **1.81** | 1.22 | 1.20 |
| Pseudo-Label | | 27.36 | 27.67 | 3.01 | 2.91 | 2.19 | 2.11 | 35.08 | 35.12 | 1.87 | 1.86 | 1.22 | 1.21 |
| MoCo | | 27.11 | 26.90 | 3.01 | 2.95 | 2.19 | 2.16 | 34.95 | 34.93 | 1.83 | 1.82 | 1.19 | 1.18 |
| MocoST | | 28.59 | 28.56 | 2.95 | 2.88 | 2.11 | 2.05 | 36.03 | 36.00 | 1.84 | 1.82 | 1.17 | 1.16 |
| CrossEntropy-H | 10 | 51.36 | 51.31 | 2.54 | 2.47 | 1.23 | 1.20 | 54.92 | 54.80 | 2.05 | 2.04 | **0.92** | **0.92** |
| PseudoLabel-H | | 51.35 | 51.28 | 2.56 | 2.49 | 1.24 | 1.21 | 55.02 | 55.02 | 2.04 | 2.04 | **0.92** | **0.92** |
| ST-H | | 40.61 | 40.65 | 2.77 | 2.66 | 1.64 | 1.58 | 46.67 | 46.60 | 1.96 | 1.94 | 1.04 | 1.03 |
| Moco-H | | 51.23 | 51.27 | 2.57 | 2.50 | 1.25 | 1.22 | **55.13** | 55.03 | 2.06 | 2.06 | **0.92** | 0.93 |
| MocoST-H | | 50.33 | 50.26 | 2.56 | 2.50 | 1.27 | 1.24 | 54.11 | 54.04 | 2.04 | 2.05 | 0.94 | 0.94 |

Table 3: Results of the proposed HiE approach on the semi-supervised setting on the *iNaturalist-19* dataset. Best results on each metric are emphasized in **bold**.

## 4.2 Semi Supervised Learning

We additionally validate our approach on fine-grained classification in the semi-supervised setting. We assume that coarse annotations are available for all the samples; however, fine-grained annotations are available only for a few samples in each class. The studied setting differs from cases [13], which divide the fine-grained classes into two categories: base classes with abundant annotated samples and novel classes where only a few annotated samples are available. This framework [13] also limits the evaluation only to the novel classes. Our setting imposes firm limits on difficult-to-obtain fine-grained labels. In the extreme setting, we assume only ten annotated training samples are available for each fine-grained class. We also evaluate the performance across all the fine-grained classes together.

We train a cross-entropy baseline exclusively using the annotated fine-grained samples. We compare its performance with representative semi-supervised methods, including Pseudo-Label [37], Self-Training (ST) with distillation [42], Self-Supervised learning (MoCo) with distillation [14] and a combination of Self-Supervised learning and Self Training (MoCo+ST). We also include a variation of all these methods (Pseudo-Label-H, MoCo-H, ST-H, MoCoST-H) with hierarchical supervised loss as described in [2] with "genus" supervision. The hierarchical supervised loss is applied to the coarse predictions obtained by marginalizing the fine-grained class predictions. The proposed hierarchical ensemble is a complementary approach, and we compare the performance of the above methods with and without applying the HiE.

The results are illustrated in Table 3. As expected, the performance of the cross-entropy baseline steeply drops with the reduction in training data. The performance of semi-supervised methods (Pseudo-Label, ST, MoCo, MoCo-ST) improves over the cross-entropy baseline; however, they fail to retain the overall performance. The semi-supervised methods become more effective with the reduced number of labeled examples (providing better proportional gains over the cross-entropy baseline). Incorporating hierarchical supervised loss brings remarkable gains in *top-1* accuracy across all experiments. For instance, in the lowest data regime (using only ten fine-grained annotated samples for training), hierarchical supervised loss brings twofold improvement over the cross-entropy baseline. The performance gains showcase the efficacy of utilizing coarse taxonomic labels in the studied setting.

Employing the proposed Hierarchical Ensemble brings consistent gains across all the methods on the three studied evaluation metrics. The combined approach of Moco-H/PseudoLabel-H with HiE

| Model | Vanilla | | | CRM | | | HiE | | |
|---|---|---|---|---|---|---|---|---|---|
| | Top 1 Acc. | Mistake Severity | Hier. Dist@1 | Top 1 Acc. | Mistake Severity | Hier. Dist@1 | Top 1 Acc. | Mistake Severity | Hier. Dist@1 |
| Mobilenet_V3 | 45.86 | 2.89 | 1.56 | 45.83 | 2.79 | 1.51 | 46.66 | 2.69 | 1.44 |
| ResNet18 | 63.63 | 2.39 | 0.87 | 63.65 | 2.31 | 0.84 | 64.65 | 2.15 | 0.76 |
| ResNet50 | 69.48 | 2.23 | 0.68 | 69.49 | 2.19 | 0.67 | 70.34 | 2.03 | 0.60 |
| ResNet101 | 70.86 | 2.14 | 0.62 | 70.87 | 2.11 | 0.61 | 71.66 | 1.96 | 0.56 |
| EfficientNet-B0 | 67.69 | 2.25 | 0.73 | 67.68 | 2.21 | 0.71 | 68.75 | 2.01 | 0.63 |
| DenseNet121 | 67.86 | 2.27 | 0.73 | 67.89 | 2.23 | 0.72 | 68.97 | 2.04 | 0.63 |
| DeiT | 64.71 | 2.38 | 0.84 | 64.74 | 2.32 | 0.82 | 65.73 | 2.14 | 0.73 |
| ViT | 66.19 | 2.36 | 0.80 | 66.27 | 2.29 | 0.77 | 67.56 | 2.06 | 0.67 |
| SwinT | 75.08 | 2.10 | 0.52 | 75.09 | 2.06 | 0.51 | 75.74 | 1.88 | 0.46 |

Table 4: Results of the proposed HiE approach with different architectures on *species* taxonomy of the *iNaturalist-19* dataset.

| Hier. Levels | Top-1 Acc. | Mistake Severity | Hier. Dist @ 1 | Hier. Dist @ 5 | Hier. Dist @ 20 |
|---|---|---|---|---|---|
| {7} | 63.63 | 2.39 | 0.87 | 1.97 | 3.25 |
| {6, 7} | 64.65 | 2.15 | 0.76 | 1.33 | 2.19 |
| {5 - 7} | **64.72** | **2.09** | **0.74** | **1.26** | 2.00 |
| {4 - 7} | 64.48 | 2.12 | 0.75 | **1.26** | 1.91 |
| {3 - 7} | 64.47 | 2.12 | 0.75 | **1.26** | **1.89** |
| {2 - 7} | 64.46 | 2.13 | 0.76 | 1.27 | **1.89** |
| {1 - 7} | 64.44 | 2.14 | 0.76 | 1.27 | **1.89** |

Table 5: Results of the proposed HiE approach on fine-grained level (*species*) of *iNaturalist-19* dataset when incorporating multiple hierarchical levels. Best results are highlighted in **bold**.

achieves a new state of the art in all settings. Using ten annotated examples for each fine-grained class, Moco-H with HiE achieves a *top-1* accuracy of 55.13, which is higher than the performance of cross entropy baselines using a hundred annotated samples per fine-grained class. The performance gains of HiE are more evident on the *avg. mistake severity* and *Hier dist@1* metrics, bringing profound gains over just using the hierarchical supervised loss. For instance, in 100 sample experiments, the *Hier dist@1* drops from 1.17 to 1.10 (6% improvement) by applying hierarchical supervised loss. In contrast, applying HiE brings 24% improvement, reducing *Hier dist@1* to 0.89.

Finally, we observe that in Table 1, a cross-entropy baseline with Resnet-18 backbone trained on the entirety of the *iNaturalist-19* dataset, i.e., 187385 examples with fine-grained labels achieves a *top-1* accuracy of 63.56%. In the semi-supervised setting using coarse-grained labels for entire data and just using 10 examples with fine-grained annotations for each class (10100 training samples), Moco-H with HiE is able to recover the significant portion of the underlying performance (achieving *top-1* accuracy of 55.13), reducing the requirement of expert annotations to only 0.5% of the total number of samples.

## 4.3 Performance across different architectures

We also showcase the generalizability of the proposed HiE approach across different vision architectures in Table 4. We utilize commonly used vision architectures like MobileNet V3 [43], Resnet variants [44], EfficientNet [45], DenseNet121 [46], and tiny variants of DeiT [47], ViT [48] and SwinT [49]. All networks are trained with the same hyper-parameter setting. We compare the proposed HiE against the vanilla and CRM-based post-hoc approach. HiE outperforms other approaches by significant margins across different architectures and on all metrics. Furthermore, similar performance gains are observed irrespective of model capacity, avg. gain of 0.9% in top-1 accuracy across resnet variants.

## 4.4 Incorporating multiple hierarchical levels

For all the above experiments, we only use two levels of hierarchies. As an additional ablation study, we validate the effectiveness of HiE in the presence of multiple hierarchical levels in Table 5. Specifically, we train a separate classifier for each hierarchical level and cascade the predictions top-down. We observe that incorporating the hierarchical information from two levels (levels 5, 6) above the leaf level gives the best performance; the gains taper off gradually as we incorporate information from higher levels. The reason for this dip in performance is that the higher levels do not have much additional information as they cover too many classes (e.g., on iNaturalist: kingdom

(level 1), phylum (level 2), and class (level 3) have 3, 4 and 9 classes respectively) which does not help much in improving the fine-grained predictions further.

## 5   Limitation

The main limitation of HiE is the requirement of training an additional network for the coarse level. However, existing approaches employ a full hierarchy with a separate classification head for each hierarchical level and additional losses to enforce the hierarchical structure, making their training process complex and difficult to converge. Instead, our approach trained with vanilla cross-entropy loss works with partial hierarchy and trade-offs the extra compute for adaptability, reproducibility and simplicity in training. A future avenue for exploration would be to learn disentangled coarse and fine-grained features imitating HiE in a unified architecture. Another limitation is the assumption regarding the availability of underlying label hierarchy. To overcome this limitation, we can exploit large-language models [50] to obtain label hierarchy using raw class labels.

## 6   Conclusion

We proposed using Hierarchical Ensembles (HiE) of independently trained networks over coarse and fine-grained levels of the label hierarchy. In terms of mistake severity, our proposed post-hoc correction consistently outperforms state-of-the-art methods in deep hierarchy-aware image classification by large margins in terms of decrease in *avg. mistake severity* and *hierarchical distance@k*, while simultaneously improving the *top-1* accuracy. In the semi-supervised paradigm, we show that HiE delivers consistent performance gains when used in conjunction with off-the-shelf semi-supervised learning algorithms. We show that, comparable performance to a fully supervised baseline can be attained, even using merely 10 annotations for each fine-grained class on a large fine-grained image classification dataset encompassing 1010 classes.

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
