# OpenReview forum: "Test-Time Amendment with a Coarse Classifier for Fine-Grained Classification"
_NeurIPS.cc/2023/Conference — NeurIPS 2023 poster_

### Official Review · Reviewer_39mn · 2023-07-04

**Soundness:** 2 fair
**Presentation:** 3 good
**Contribution:** 2 fair
**Rating:** 4
**Confidence:** 4

**Summary:**

The paper considers the problem of fine-grained classification and proposes to use label hierarchy information at test time to improve the performance of the fine-grained classifier. The overarching goal is to improve the top-1 accuracy while at the same time reducing the severity of the mistakes (e.g., misclassifying species from the same kind or family is more acceptable than making a totally unrelated prediction).

The paper presents a post-hoc correction technique called Hierarchical Ensembles (HiE). The key idea is to train 2 classifiers - 1 fine-grained and 1 coarse-grained. At inference time, the predictions of the coarse-grained classifier are used to re-weight the probabilities of the fine-grained one. The authors perform experiments on 2 standard benchmark datasets: iNaturalist-19 and tieredImageNet-H and claim to achieve new state-of-the-art results on both of them. They also show the promise of their method in a semi supervised learning setup where the coarse grained labels are known for all samples in the dataset but the fine-grained (being more expensive to collect) only for few samples per class.

**Strengths:**

* The paper is written well and is easy to read.
* The paper extends and builds up on prior works and seems to exceed their results.
* The Semi Supervised Learning setup is interesting.
* The conducted experiments and ablations are sensible.

**Weaknesses:**

My main concern is with the methodology (Section 3). I understand the final decision rule (Eqs. (2) and (3)) but its derivation and motivation is confusing to me.

L118: "Assuming conditional independence between the estimated logits...": $i ⫫ i_{parent} \mid x$ means $P(i, i_{parent} \mid x) = P(i \mid x)P(i_{parent} \mid x)$. Thus, to the best of my knowledge, the denominator in Eq. (1) should not be there. Moreover, I believe that the conditional independence assumption is unrealistic for the considered setup. It does not feel right that the fine-grained prediction/probability is independent of its parent. This affects the derivations in the rest of the section.

Minor:

* The following related, in my opinion, citation is missing:\
Ridnik et al., ImageNet-21K Pretraining for the Masses, NeurIPS 2021 (Datasets and Benchmarks)\
They train ImageNet-21K classifiers using the WordNet hierarchy.
* For deep hierarchies, it may still be costly to collect labels from the penultimate hierarchy level. It would be useful if the authors include the set of labels from the last two levels for each of the datasets and refer to them from the main paper.

**Questions:**

Q1: Could you please clarify how you formally derive the intuition for the decision rule defined in Eqs. (2) and (3)?

Q2: Would it help if you use the logits from $\hat{y}_{L}$ and

$\hat{y}_{L-1}$ directly instead of the softmaxed probabilities in $Q$ and $R$?

Q3: To complete the discussion from L143-147, you can also consider the case when the fine-grained prediction is correct but the coarse-level prediction is wrong.

Q4: I am curious how often there is a mismatch between the fine-grained and the coarse classifier. And what is the distribution / the average of the depth of their LCA nodes.

Q5: L220-221: "... training a separate network at the coarse label allows explicit disentanglement of features at the coarse and fine-grained levels ...". I don't believe that the current set of experiments is sufficient to make such a strong claim.

Q6: Does the $\pm$ in the tables refer to standard deviation? How many runs were executed?

Q7: Sec. 4.4: Following the comments in the Weaknesses section above, it might be useful to formally clarify what is meant by "cascade the predictions top-down" (L279-280). Also, do you need to incorporate consecutive hierarchical levels (e.g., would it be possible and sufficient to take levels 5 and 7, omitting level 6)?

Q8: Sec 5: What do preliminary experiments suggest if you unify the architecture of the fine-grained and the coarse-grained classifier (and, e.g., only fine-tune separate classification heads for the two classifiers)?

---

> ### Author Rebuttal · Authors · 2023-08-08
>
> Thank you for your constructive and thoughtful comments. They were indeed helpful to improve the paper. We take this opportunity to address your concerns:
>
> * **Q1: Clarify how you formally derive the intuition for the decision rule defined in Eqs. (2) and (3)?** We agree with your comments and we recognize that bypassing the intermediary thinking process resulted in a lack of clarity. We carefully redo the derivation of the proposed decision rule in the global response above (at the top). Furthermore, we would like to point out that if we simply consider Eqn1 as a modified scoring function, the proof and the strong empirical results still hold.
>
> * **Q2: Would it help if you use the logits?** Using logits instead of softmax probabilities for HiE improves over the baseline, however, the improvement is lower compared to that using softmax. We provide the comparison below.
> | | Top-1 | Avg. Mist | Hie Dist@1 | Hie Dist@5 | Hie Dist@20 |
> |:-------:|:-----:|:---------:|:----------:|:----------:|:-----------:|
> | Logits | 64.46 | 2.16 | 0.77 | 1.35 | 2.43 |
> | Softmax | 64.61 | 2.15 | 0.76 | 1.33 | 2.19 |
>
> * **Q3: L143-147 Consider the case when the fine-grained prediction is correct but the coarse-level prediction is wrong** We provide the mistake mismatch distribution on the iNaturalist dataset below. There are a few examples where fine-grained predictions are correct, while coarse-grained predictions are wrong (3.46%). In such cases, HiE may hamper the performance. However, that is only a small fraction compared to the more interesting scenario of coarse-correct and fine-incorrect (25.19%). We will add this discussion to the paper.
> | Before HiE | Fine Correct | Fine Incorrect |
> |------------------|--------------|----------------|
> | Coarse Correct | 60.24 | 25.19  |
> | Coarse Incorrect | 3.46 | 11.11 |
>
> * **Q4: Mismatch between the fine-grained and the coarse classifier. And what is the distribution / the average of the depth of their LCA nodes** We provide the distribution in response to Q3. We provide the average LCA, between the predicted fine-grained and coarse-grained labels below (for each of the four cases)
> | Before HiE | Fine Correct | Fine Incorrect |
> |------------------|--------------|----------------|
> | Coarse Correct |  1.00 | 1.65 |
> | Coarse Incorrect | 4.07 | 3.07 |
>
> * **Q5: L220-221: "training a separate network at the coarse label allows explicit disentanglement of features at the coarse and fine-grained levels"** We thank the reviewer for pointing this out. Indeed, we claim that the two models learn complementary features which can be combined using our HiE method. We will update the paper to avoid using the term explicit disentanglement.
>
> * **Q6: Does the ± in the tables refer to standard deviation? How many runs were executed?** Yes, it indicates the standard deviation over 5 runs.
>
> * **Q7: Sec. 4.4: Cascade and skipping Hierarchy** In the Cascade process, we apply HiE on level 2 using level 1 for coarse predictions. The updated level 2 predictions are then used as coarse predictions to apply HiE on level 3. The process repeats till the leaf node. \
> Our approach does not necessarily require consecutive hierarchical levels. HiE results on level 7 of the iNaturalist dataset, directly using level 5 (omitting level 6) for coarse prediction are given below. It leads to noticeable gains over the baseline, however, the performance remains marginally lower compared to using level 6 for coarse classification.
> | Method | Top-1 | Avg. Mistakes | Hier Dist@1 | Hier Dist@5 | Hier Dist@20 |
> |------------|------:|--------------:|------------:|------------:|-------------:|
> | Baseline | 63.70 | 2.38 | 0.86 | 1.96 | 3.24 |
> | HiE (Level 6) | 64.61 | 2.15 | 0.76 | 1.33 | 2.19 |
> | HiE (Level 5) | 64.49 | 2.16 | 0.77 | 1.36 | 2.09 |
>
> * **Q8: Sec 5 Preliminary experiments on unified architecture** Compared to the baseline, unified architecture with a common backbone for feature extraction and separate classification heads for coarse and fine-grained level results in decreased performance for fine-grained classification (63.32% vs 63.70%) and improved performance on coarse classification (86.15% vs 85.30%). Applying HiE between the two classification heads improves the hierarchical metrics but results in lower top-1 accuracy.
> | Method | Top-1 | Avg. Mistakes | Hier Dist@1 | Hier Dist@5 | Hier Dist@20 |
> |------------|------:|--------------:|------------:|------------:|-------------:|
> | Baseline | 63.70 | 2.38 | 0.86 | 1.96 | 3.24 |
> | Unified Arch. | 63.32 | 2.20 | 0.81 | 1.50 | 2.62 |
> | HiE Unified| 63.10 | 2.17 | 0.80 | 1.33 | 2.26 |
> | HiE Separate | 64.61 | 2.15 | 0.76 | 1.33 | 2.19 |
>
> **Missing Citation:** Thank you for pointing out this reference. We will add this to the paper. We would also like to point out that the method in [Ridnik et al., ImageNet-21K Pretraining for the Masses] is similar to [25], which we compare against in our experiments.

---

> > ### Comment · Reviewer_39mn · 2023-08-17
> >
> > Thank you, increased my score to 4 in light of the further details provided by the authors and will continue the discussion in the main response.

---

### Official Review · Reviewer_yfvr · 2023-07-05

**Soundness:** 3 good
**Presentation:** 3 good
**Contribution:** 3 good
**Rating:** 6
**Confidence:** 3

**Summary:**

This method introduces a novel approach to achieve state-of-the-art fine-grained image classification. It addresses the issue of mistake severity by developing the Hierarchical Ensemble (HiE) loss, which effectively penalizes incorrect predictions of both course and fine-grained labels. The HiE loss combines the probabilities of predicting the course label and the fine-grained label, providing a joint probability measure.

The authors of this method provide a proof that accurately predicting the course label probability enhances overall accuracy. To validate their approach, they conducted extensive experimentation on two hierarchical datasets, comparing their method against various existing related works and three additional baselines. The evaluation metrics used include top-1 error, mistake severity, and hierarchical distance. The results clearly demonstrate that their method outperforms comparable approaches across all these measures.
Moreover, the authors conducted further experiments in a semi-supervised setting to assess the performance of their method when only 10% of labels are available. Remarkably, their approach exhibits significant performance improvements even under such limited label availability, highlighting its effectiveness and robustness.

In addition to the aforementioned experiments, the authors explored the impact of using different pretrained backbone models in their method. They conducted experiments with various hierarchical depths to evaluate the method's adaptability and generalization across different classification hierarchies. The results of these additional experiments further support the superiority and versatility of their proposed method.

Overall, this research presents a compelling method for fine-grained image classification, showcasing its state-of-the-art performance, reduced mistake severity, and its ability to improve accuracy even with limited labeled data. The comprehensive experimentation and analysis conducted by the authors demonstrate the effectiveness and versatility of their approach in various scenarios, making it a valuable contribution to the field of image classification.


**Strengths:**

The text is very well written and easy to understand. It’s very clear what the problem is and how their method can be used to increase performance.

The authors conducted a comprehensive range of experiments to thoroughly evaluate the performance of their model.

The method demonstrates robustness by effectively utilizing any pretrained backbone model.

**Weaknesses:**

Since I’m not as familiar with this method, it would have been helpful to have a clearer description of these metrics within the paper, as I had to refer to the related works to gain a better understanding.

**Questions:**




**Limitations:**

I believe the authors appropriately discussed the limitations of their method.

The collection of course-grained labels in addition to fine-grained labels seems very costly but this is outside of the scope of this method.

---

> ### Author Rebuttal · Authors · 2023-08-08
>
> We would like to thank the reviewer for their positive feedback.
>
> **Evaluation Metrics:** Thank you for the suggestion. We will update the paper to discuss the evaluation metrics in more detail and make it self-contained.

---

### Official Review · Reviewer_tPTQ · 2023-07-07

**Soundness:** 4 excellent
**Presentation:** 3 good
**Contribution:** 3 good
**Rating:** 5
**Confidence:** 3

**Summary:**

This paper proposes a novel approach called Hierarchical Ensembles (HiE) to improve the performance of fine-grained classification by utilizing a label hierarchy and coarse-grained predictions at test-time. The method significantly reduces mistake severity while improving top-1 accuracy on benchmark datasets, achieving state-of-the-art results. The approach is also effective in the semi-supervised setting, bringing notable gains in accuracy and reducing mistake severity as training data decreases for fine-grained classes.

**Strengths:**

	Originality: The paper introduces a novel approach called Hierarchical Ensembles (HiE) that combines coarse-grained predictions and label hierarchy to improve the performance of fine-grained classification. This approach is unique and addresses the challenge of reducing mistake severity while improving accuracy in fine-grained classification. Therefore, the paper demonstrates originality in its proposed methodology.

	Quality: The paper achieves state-of-the-art results on benchmark datasets by significantly reducing mistake severity and improving top-1 accuracy. The approach is effective not only in the supervised setting but also in the semi-supervised setting, bringing notable gains in accuracy. The paper also compares its approach with previous baselines and demonstrates superior performance. These factors indicate the high quality of the research presented in the paper.

	Clarity: The paper provides a clear and concise explanation of the proposed approach, including the motivation, methodology, and experimental results. The authors effectively communicate the problem statement, the significance of their approach, and the experimental setup. The paper also includes figures and examples to enhance clarity. Overall, the paper is well-written and easy to understand.

	Significance: The paper addresses the challenge of fine-grained classification, which requires domain expertise and large amounts of labeled data. By utilizing coarse-grained predictions and label hierarchy, the proposed approach significantly reduces mistake severity and improves accuracy. This has practical implications in various domains where fine-grained classification is important, such as image recognition and object detection. The paper's state-of-the-art results and compatibility with existing semi-supervised methods further highlight its significance.


**Weaknesses:**

There are some problems, which must be solved before it is considered for publication. If the following problems are well-addressed, this reviewer believes that the essential contribution of this paper are important for fine-grained classification. The paper has some context inconsistency errors, for example, it refers to these semi-supervised models on line 233, but then refers to them as self-supervised models on lines 242 and 244. In addition, although the complementary method proposed in this paper can be applied to the off-the-shelf model, the overall innovation of the paper is insufficient.

**Questions:**

This paper makes extensive and comprehensive experiments on the new method proposed by the author, which fully proves the effectiveness of the method. The innovations of the paper can be applied to many off-the-shelf models, but there are not enough of them. If more general structures could be proposed, this paper would be able to make a greater contribution to the field dealt with.

**Limitations:**

The authors adequately address their proposed limitations in previous models, as their proposed method is a complementary structure that can be applied to any existing model and can effectively improve the performance of the model on the dataset.

---

> ### Author Rebuttal · Authors · 2023-08-08
>
> We thank the reviewer for their thoughtful feedback and would like to address their concerns below.
>
> **Typo on lines 242-244:** Thank you for pointing out this inconsistency. We will update the paper to fix this. It should be semi-supervised on lines 242 and 244.
>
> **Overall Innovation of the work:** We would like to point out that the key innovation in our work is proposing a theoretically principled method that can be applied off-the-shelf to a variety of models to improve the accuracy as well as reduce the severity of mistakes in the setting of hierarchical classification. We present comprehensive experiments, study and compare against several methods exploring hierarchical architectures, hierarchical loss functions and hierarchical embeddings. We find that, albeit minimal, the proposed approach achieves state-of-the-art performance, while providing advantages in terms of adaptability, reproducibility and simplicity of training. We believe that these findings are of significant value to the community.

---

### Official Review · Reviewer_cuDf · 2023-07-10

**Soundness:** 3 good
**Presentation:** 2 fair
**Contribution:** 2 fair
**Rating:** 5
**Confidence:** 4

**Summary:**

This paper focuses on label hierarchy problems and proposes using Hierarchical Ensembles (HiE) of independently trained networks over coarse and fine-grained levels. The reported experimental results show that the proposed method can achieve comparable performance to a fully supervised baseline, even using merely 10 annotations for each fine-grained class on a large fine-grained image classification dataset encompassing 1010 classes.

**Strengths:**

1. The topic of this paper is well-introduced and the proposed approach is straightforward.

2. The proposed HiE utilizes label hierarchy to improve the performance of fine-grained classification at test time using coarse-grained predictions.

3. The proposed methods and experimental results are well presented and the manuscript is overall well organized.

4. Performance of the proposed method is promising according to the comparison experiments with other state-of-the-art studies.


**Weaknesses:**

1. The paper lacks a general investigation of hierarchical architectures and hierarchical embeddings.

2. The proof of the Theorem 3.1 is not convincing, thus making the overall scheme lack theoretical support.

3. The experimental setup of semi-supervised learning is not shown clearly, so it is not possible to judge whether the comparison of the proposed method with other methods is fair.

4. Some errors, such as why Figure 2 is 4 levels.


**Questions:**

Please give a detailed motivation for this paper and a detailed derivation of Theorem 3.1

**Limitations:**

The subject of the paper is Test-Time Amendment, thus limiting the potential performance improvement.

---

> ### Author Rebuttal · Authors · 2023-08-08
>
> We thank the reviewer for their positive feedback. We address the reviewer's concerns and questions below:
>
> **Motivation of the paper:** The primary motivation behind the paper is that training independent coarse and fine-grained help to learn complimentary features. We provide the thought process behind Eqn1 in the global response above (at the top). We show that a minimal post-hoc approach improves upon complex efforts exploring hierarchical architectures, hierarchical loss functions and hierarchical embeddings on two important problems of reducing mistake severity and semi-supervised learning. The proposed approach also provides significant advantages in terms of adaptability, reproducibility and simplicity of training.
>
> **Detailed derivation of theorem:**  The detailed derivation of theorem 3.1 is provided below.
>
> We show that if we make a correct prediction at the coarse level, the proposed Hierarchical Ensemble (HiE) is guaranteed to improve the downstream predictions at the fine-grained classification task.
>
> **Theorem 3.1:** Assuming, $Q = [q_1, q_2 , ..., q_{N_L}]$ and  $R = [r_1, r_2 , ..., r_{N_{L-1}}]$ are the predictions obtained at the fine and coarse grained labels for a given input $x$, such that $\sum_{i=1}^{N_L} q_i = 1$ and $\sum_{i=1}^{N_{L-1}} r_i = 1$. Let $g$ and $g_{parent}$ be the ground truth labels at the fine grained and the coarse grained levels respectively.
>
> Now assuming that the coarse label is correctly predicted by the coarse prediction network i.e. $\mathrm{argmax} (R) = g_{parent}$, we wish to prove that:
> $$
> \frac{q_g \mbox{ . } r_{g_{parent}}}{\sum_{j=1}^{N_L} q_j \mbox{ . } r_{j_{parent}}} \ge q_g    \quad (1)
> $$
>
> Where L.H.S is the prediction for the ground truth class using HiE and R.H.S is the direct prediction of the fine grained network for the ground truth class.
>
> **Proof:** The denominator iterates over the fine grained predictions and multiplies them with their parent's prediction score. This is equivalent to iterating over the coarse label predictions and multiplying with the sum of prediction scores for all its children. By rewriting the denominator, we obtain:
>
> $$
> \sum_{j=1}^{N_L} q_j \mbox{ . } r_{j_{parent}} = \sum_{j=1}^{N_{L-1}} r_j \sum_{i\in j_{child}} q_{i}  \quad (2)
> $$
>
> Assuming, $\sum_{i\in j_{child}} q_{i} = z_j$
> $$
> \sum_{j=1}^{N_{L-1}} r_j \sum_{i\in j_{child}} q_{i} = \sum_{j=1}^{N_{L-1}}  r_j \mbox{ . } z_j = R^T Z       \quad (3)
> $$
>
> Invoking Holder's inequality, using $a=\infty$ and $b=1$, ($1/a + 1/b = 1$), we obtain:
> $$
> R^T Z \le \lVert R \rVert_{\infty}  \lVert Z \rVert_{1}      \quad (4)
> $$
>
> Since, $\lVert R \rVert_{\infty} = \mathrm{max} (R) = r_{g_{parent}}$ and $\lVert Z \rVert_{1}  = \sum_{i=1}^{N_L} q_i = 1$. We can say:
> $$
> R^T Z  \le \lVert R \rVert_{\infty} \lVert Z \rVert_{1}  \le r_{g_{parent}}   \quad (5)
> $$
>
> Given Eqn 5, we can conclude that:
> $$
> \frac{r_{g_{parent}}}{\sum_{j=1}^{N_L} q_j \mbox{ . } r_{j_{parent}}} \ge 1   \implies  \frac{q_g \mbox{ . } r_{g_{parent}}}{\sum_{j=1}^{N_L} q_j \mbox{ . } r_{j_{parent}}} \ge q_g \quad (6)
> $$
>
>
> **Hierarchical Architectures and Embeddings:** We would like to point out that we sufficiently discuss hierarchical architectures and embeddings in the related work. We compare against four hierarchical architectures [8,11,25,6] and a hierarchical embeddings-based method [5]. We also reproduce two baselines (Cross-Entropy-H and HiE Self). \
> Additionally, based on the suggestions provided, we experiment with a unified architecture with a shared feature backbone and only different classification heads for coarse and fine-grained classes. We also experiment with applying HiE over the predictions from the separate classification heads. The results are given below:
> | Method | Top-1 | Avg. Mistakes | Hier Dist@1 | Hier Dist@5 | Hier Dist@20 |
> |------------|------:|--------------:|------------:|------------:|-------------:|
> | Baseline | 63.70 | 2.38 | 0.86 | 1.96 | 3.24 |
> | Unified Arch. | 63.32 | 2.20 | 0.81 | 1.50 | 2.62 |
> | HiE Unified| 63.10 | 2.17 | 0.80 | 1.33 | 2.26 |
> | HiE Separate | 64.61 | 2.15 | 0.76 | 1.33 | 2.19 |
>
> However, if the reviewer has additional suggestions for baselines/methods proposing hierarchical architectures and embeddings, we would be happy to incorporate it.
>
> **Clarification of the semi-supervised learning setting:** In the semi-supervised setting (section 4.2, lines 225-228), we assume the availability of a lot of coarsely labelled samples and a small number of fine-grained samples. We experiment with reducing the number of fine-grained annotations, from 100 annotations per class to merely 10 annotations per class. We compare against several methods and show results with-without CRM and with-without HiE.
>
> **Figure 2 caption:** We would like to clarify that Figure 2 refers to a 2 level hierarchy with 4 leaf classes.

---

### Author Rebuttal · Authors · 2023-08-09

We describe the steps clarifying the derivation of Eqn 1 in the paper. We use slightly different notations for the sake of improved clarity.

>          (X)
>         /   \
>        /     \
>       /       \
>     (C)       (F)
>     Graphical model for separate classifiers trained on coarse and fine-grained labels.
>     (X) is the input image, (C) and (F) are coarse and fine-grained labels, respectively.

Considering the given graphical model, by the product rule, we have:
$$
P(C, F, X) = P(C, F | X) \cdot P(X)   \quad  (1)
$$

By factorizing the graphical model we obtain,
$$
P(C, F, X) = P(C | X, \phi) \cdot P(F | X, \theta) \cdot P(X)  \quad  (2)
$$

From Eq. (1) and Eq. (2), we have,
$$
P(C, F | X) = P(C | X, \phi) \cdot P(F | X, \theta)   \quad  (3)
$$

From Eq. (3), we have conditional independence between the predictions of coarse and fine-grained classifiers, i.e.: $C \mathrel{\unicode{x2AEB}} F | X.$


Now, assuming access to a label hierarchy $\mathcal{H}$ between coarse and fine labels, we define the following score function between the coarse classifier's prediction for the $j^{th}$ coarse class $c_j$ and the fine classifier's prediction for the $i^{th}$ fine class $f_i$:
$$
S(f_i, c_j ; x ,\theta, \phi, \mathcal{H}) = P(f_i | x, \theta) \cdot P(c_j | x, \phi) \cdot \mathbb{1}_{\mathcal{H}}(c_j = \text{parent}(f_i))   \quad  (4)
$$

Where $\mathbb{1}\_{\mathcal{H}}$ is the indicator function manifesting the label hierarchy. Assuming that each fine-grained label is assigned only a single coarse label and $c_{i_{\text{parent}}}$ = $\text{parent}(f_i)$, we can simplify the above equation to:
$$
S(f_i, c_{i_{\text{parent}}} ; x ,\theta, \phi) = P(f_i | x, \theta) \cdot P(c_{i_{\text{parent}}} | x, \phi) , \text{ for } i=1,2,...,N_L,    \quad  (5)
$$

We normalize Eq. (5) to make it a valid probability density function:
$$
P(f_i, c_{i_{\text{parent}}} | x ,\theta, \phi) = \frac{P(f_i | x, \theta) \cdot P(c_{i_{\text{parent}}} | x, \phi)}{\sum_{j=1}^{N_{L}} P(f_j | x, \theta) \cdot P(c_{j_{\text{parent}}} | x, \phi)} , \text{ for } i=1,2,...,N_L,    \quad  (6)
$$

Eqn (6) is equivalent to Eqn 1 in the paper.

---

> ### Comment · Reviewer_39mn · 2023-08-18
>
> Thank you for the detailed explanation of the reasoning. I get what you are doing, but there are still some minor issues. Namely, the normalized "probability" $P$ from (6) is different from that in (1), (2) and (3). From your initial assumptions, if I am not mistaken, it must follow that $P(C = c_{i_{parent}}, F = f_i | X) = P(C = c_{i_{parent}} | X, \phi) \cdot P(F = f_i | X, \theta)$. This is what you must get when you instantiate (3) with $c_{i_{parent}}$ and $f_i$, which clearly contradicts the result in equation (6).
>
> I recommend describing (6) as a "normalized score". If we think of it as a probability, it comes from a tweaked probabilistic model that is different from the one that gives rise to equations (1)-(3). And the description of this tweaked model gets more complicated.
>
> Moreover, the leaf $i$ should be the only parameter of this normalized score (assuming $x$, $\theta$, $\phi$ and the hierarchy $\mathcal{H}$ are given). $i_{parent}$ is uniquely defined by $i$ (and $\mathcal{H}$). Therefore, your method can indeed be thought as reweighing the score (or the probability) of the leaves, given the probabilities from the coarse-grained classifier.

---

> > ### Author Response · Authors · 2023-08-18
> >
> > Thank you for your response. You are right, indeed Eqn 3 is different from Eqn 6. Eqn 3 is a joint probability matrix of dimension ($N_L$ x $N_{L-1}$), and Eqn4 only captures one value per row of this matrix (corresponding to its parent). We normalize that to obtain Eqn6.
> >
> > In the global response, we have carefully termed Eqn4 as a scoring function (in the same spirit to your suggestion of calling it a normalised score). We would avoid calling it a probability in the main paper as well.
> >
> > We also agree that it is cleaner to interpret our method in terms of reweighing the score (or the probability) of the leaves, given the probabilities from the coarse-grained classifier.
> >
> > We will update Eqn1 in the original paper as following:
> >
> > $P(i | x, \theta, \phi, H) = RHS$ , where $i_{parent} = H(i)$

---

### Decision · Program_Chairs · 2023-09-21

**Decision:**

Accept (poster)

**Comment:**

The paper proposes an approach called Hierarchical Ensembles (HiE) that utilizes label hierarchy and coarse-grained predictions to improve the performance of fine-grained classification at test-time. The paper claims to achieve state-of-the-art results on two benchmark datasets, iNaturalist-19 and tieredImageNet-H. The main concern was about the derivation of the method described in Section 3. Reviewer 39mn pointed out this issue and discussed with authors during the rebuttal phase and the authors provided a detailed explanation of their reasoning. Overall, based on the reviews and comments, I recommend an ACCEPT to this paper. The paper demonstrates its superiority over previous baselines on two benchmark datasets and also shows its applicability in the semi-supervised setting. The authors addressed most of the concerns and questions raised by the reviewers in their rebuttal, but they should also incorporate their clarifications and corrections in the final version.